# Peer review of "Epistemic uncertainties and natural hazard risk assessment – Part 2: Different natural hazard areas"

_Natural Hazards and Earth System Sciences, 2015_

## Short Comment (SC1) · 15 Feb 2016

I've read your manuscript and I've found it very interesting. It provides interesting food for thoughts. The question of epistemic uncertainties and natural hazard risk assessment and forecasting is still an open problem.

Since you included a part related to rainfall induced landslides and epistemic uncertainties (page 15, line 10-21), I would like to suggest a paper I have co-authored: Gariano et al., 2105.

Our work deals with the epistemic uncertainty due to the lack of information on landslide occurrence, in a validation procedure of regional rainfall thresholds. We have investigated the consequences of the lack of information on the contingency table and the related skill scores usually used to evaluate the forecasting performance of a thresh-

old in an early warning system. We found that even a small (1%) underestimation in the number of the considered landslides could result in a significant decrease in the performance of the system.

I hope it can be useful for your work.

Reference:

Gariano S.L., Brunetti M.T., Iovine G., Melillo M., Peruccacci S., Terranova O.G., Vennari C., Guzzetti F. (2015), Calibration and validation of rainfall thresholds for shallow landslide forecating in Sicily, Southern Italy, Geomorphology, 228, 653-665, doi:10.1007/s11069-014-1129-0.

---

## Referee Comment (RC1) · Anonymous Referee #1 · 16 Feb 2016

Manuscript # 2015_295 entitled 'Epistemic uncertainties and natural hazard risk assessment. 2. Different natural hazard areas'.

This aim of this paper is to characterise the epistemic uncertainties associated with forecasting different natural hazards. The different analysis methods in each field are compared and similarities and differences highlighted.

As suggested by the editor I have focused specifically on the section dealing with volcanic eruptions and ash clouds (section 8). Overall, this section contains a useful description of the current uncertainty analysis being undertaken in the field of volcanic ash dispersion. However, the structure of the section is confusing in parts and several sources of epistemic uncertainty are missing from the description. Finally, explicit links to the uncertainty analysis types described in the paper's conclusions would help to

link this section more closely to the overall aim of the paper. This is an ambitious and important paper that is worthy of publication.

Comments 1. Page 34, lines 6-11: The authors are rather dismissive about 'other types of observations'. Whilst there are obvious limitations associated with these low temporal or spatial resolution observations, they are often invaluable in measuring some of the physical properties of ash and ash cloud geometry that satellite retrievals currently need to make assumptions about in order to make concentration estimates. For example, ash layer depth, ash cloud height and particle size distribution. It might be better to acknowledge that combining these observations with the satellite observations can help to reduce the epistemic uncertainty?

2. Page 33, line 18: I think '. . . in order to make estimates on other physical properties such as ash column loading . . .' should be '. . .in order to make estimates of other physical . . .'

3. Page 34, lines 3-5: The authors refer to 'simulations of satellite imagery using a radiative transfer model'. This is a bit ambiguous. I assume they are referring to volcanic ash satellite retrieval schemes, but another interpretation could refer to simulations of satellite imagery using VATD output, such as performed by Millington et al. (2015). This should be stated more clearly. More generally, 'the model' is also ambiguous as there are several models in this section. Millington, S. C., R. W. Saunders, P. N. Francis, and H. N. Webster (2012), Simulated volcanic ash imagery: A method to compare NAME ash concentration forecasts with SEVIRI imagery for the Eyjafjallajökull eruption in 2010, J. Geophys. Res., 117, D00U17, doi:10.1029/2011JD016770.

4. Page 34, line 13: 'modelling of the hazard is, however, a problem of forecasting'. Since modelling in the previous 3 paragraphs was referring to volcanic ash satellite radiative transfer models, I was initially confused by this sentence. As the next few paragraphs go on to discuss VATD models, however, I assume modelling here refers to VATD models. It would be a good idea to clarify that from that point onwards the
authors are discussing dispersion models.

5. Page 34: When discussing parametric uncertainty, the authors focus almost exclusively on of the eruption source parameters (PSD, ash density, MER). I was surprised that uncertainty associated with the input meteorological fields was not discussed. Was this left out deliberately? If so, why?

6. Page 35, lines 5-18: The structural uncertainties associated with missing near-source processes (bent plume, ash aggregation, effect of gravity currents) are dispersed through p34 and p35. Describing these missing processes in a separate paragraph might help with the clarity of the section.

7. Page 35, line 20: The authors refer to the 'dispersion parameters' but there is no description of what these are, or any of the other parameter uncertainties associated with processes that are represented in VATD models, such as wet and dry deposition, sedimentation, turbulence etc. Why is there no discussion of parameter uncertainties?

8. Page 35: In the conclusions section the authors nicely split uncertainty analysis into 3 types. The parametric sensitivity studies described on page 35 all fall under the forward analysis approach. It might make it easier for the reader to link to the conclusions section if the authors could refer to the analysis type here.

9. Page 36, lines 9-14. Similarly, how does the method of Denlinger et al. (2012) fit into the 3 types of uncertainty analysis framework? Is this a source inversion analysis?

10. Page 37, line 1-2. Given the operational time constraints, do the authors think that statistical emulators have a role to play in characterising volcanic ash forecasting uncertainties?

---

## Referee Comment (RC2) · Anonymous Referee #2 · 22 Feb 2016

I was asked to review the document primarily from a seismic hazard perspective.

GENERAL COMMENTS

This is a very long, yet incomplete, review paper of epistemic uncertainties associated with different natural hazards. I wanted to be reading about novel solutions to the technically challenging problems highlighted. Perhaps a better format would be to include these descriptions of the epistemic uncertainties within papers that specifically address them?

It was not evident that this review added much more than, for example, the book on "Risk and uncertainty for Natural Hazards" edited by Rougier, Sparks and Hill.

Given the lack of new material - the quality of the paper rested on whether the discus-

sion of epistemic uncertainties was thought provoking and nuanced. For me, this was also lacking.

My recommendation to reject is based on the lack of new content, a lack of addressing the issues posed and the length of the paper.

SPECIFIC COMMENTS

I give two specific examples of the paper's limitations within the seismic hazard section below, though there are more...:

Paragraph starting 815: This paragraph starts by introducing the problems of the long-term occurrence rates of extreme or "characteristic" earthquakes and then goes on to talk about maximum magnitudes.

Further paragraph 846 talks about "non-Poisson and quite-periodic" events at subduction zones.

The language implicit asserts that large earthquake are at least semi-periodic since the language used talks about recurrence rates and characteristic events. Both of these are debated issues which are, only potentially applicable to certain regions - only the latter paragraph makes this clear. Do the authors intend this discussion to be restricted to subduction zones or a wider coverage? For example, how do the Sichuan (Mw 7.9), New Madrid (8.1?), Gujarat (Mw 7.7) and Parkfield earthquakes fit into this framework?

Further, assuming a characteristic model is a potentially large source of epistemic uncertainty as there is not consensus about either the definition or validity of the characteristic model.

Part of the challenge, is that records of seismic data are insufficiently short. An aleatoric reason for this is that the moments of the Gutenberg Richter relation are not finite. Measures of longer-term deformation are required to close this distribution. However, what will not resolve this issue (as implied by paragraph starting on line 815), is improving the completeness threshold or reliability of modern earthquake catalogues. Both of which are highly desirable, but will not contain the evidence to clinch long term recurrence rates of evidence for characteristic events, in my opinion. Hence, the second half of the paragraph does not address the 2 challenges identified in the first half.

Paragraph starting 833 The inclusion of geodetic information will undoubtably help refine estimates.

Statistically, one of the problems with estimation of the maximum magnitude is that our evidence for it can only ever show us to have underestimated it. Since the geological process is one of the gradual destruction of evidence – we need to recognise large uncertainty in its evaluation. It would have been good to have show how to consider these uncertainties within the geodetic case.

Further, it seems unlikely we will be able to apply it globally and uniformly due to resource limitations. Since we generally go to look at regions we believe are scientifically interesting, we are biased at looking at regions which have a signal. This bias means we can systematically miss swathes of hazardous regions.

A good example of where we are developing new constraints is seafloor geodesy.

INCOMPLETENESS OF DISCUSSION

There are also several examples of epistemic discussions missing from the paper.

For example, how should one construct the background seismic hazard away from known active regions - Christchurch is a good example of this where the hazard was underestimated. Is there a prospect for PSHA to be able to inform those communities better in the future or was the earthquake and subsequent revision of building codes PSHA performing "well"?

In a related matter, quantifying/mapping of what is measurable in principle also introduces systematic model uncertainties. Being sat on a modern sedimentary basin, the structure underlying the Christchurch region was masked. Similarly, a large proportion

of earthquakes occur on unmapped structures. Geodetics can map slowly deforming structures – but not all unmapped structures are slowly deforming – they can remain locked until large earthquakes or similar change that situation.

A further source of bias in the analysis of seismic hazard remains poor application of basic statistics. Many papers still apply least squares fitting to find the b-value – a process that was identified as being severely biased in the 1960s and yet is still widely applied. This application of using an inappropriate error structure and underestimating uncertainty at large magnitudes is certainly a classic example of applying the wrong model (i.e. epistemic).

---

## Editor Comment (EC1) · R E Chandler (Editor) · 13 Mar 2016

I was a little surprised to see that the Copernicus editorial system has automatically closed the discussion on this paper, because there are several other reviewers who haven't submitted their reports. I wanted a range of views from different hazard specialists, but only a couple of the areas are represented. Nonetheless, the comments received so far on this and on the companion paper are fairly consistent and in accordance with my own assessment, so it probably does no harm to return this one to the authors for a *very substantial* revision at this point. I'm afraid it needs a lot of work, if it is to fulfil its promise of a wide-ranging review in a special issue devoted to state-of-the-art treatment of uncertainty. Similar comments will come back on the companion paper in due course; I am still waiting for a final referee report on that one however, and I want to see that report before finalising my own comments.

The present paper sets out to review the ways in which generic issues of epistemic uncertainty, set out in the companion paper, have been addressed in different natural hazard areas. It inevitably inherits the perspective of the companion paper, so some of my major concerns about that paper carry through to this one as well. One of the most serious concerns is that both papers need to provide a much more balanced perspective: at present several techniques are criticised inappropriately, in a way that confuses the underlying concept with the way in which the techniques are often applied by non-specialists. I will say more about this in my report on the companion paper itself. However, for the present paper as well there are a few places where assertions need to be reframed to present a more balanced case.

An obvious general comment about the present paper is that it is very long. There is inevitably some overlap with the companion paper, but there are also a couple of places where essentially the same material is repeated and it would be good to identify opportunities to reduce the length.

A final general comment is that the treatment of the different hazard areas is rather uneven: for some areas the paper provides a reasonably comprehensive review of the landscape, but for others I have the sense that it does not so much provide a comprehensive review of the relevant issues as promote the relevant authors' own work. This is not really appropriate. The material on pyroclastic flows stands out particularly in this respect (although it is not the only offender): this reads less as a a comprehensive review of uncertainty-relevant issues than as a list of gaps in our understanding of Vesuvius.

Specific comments are as follows:

- Lines 25–28 "In each case it is common practice ... underestimation of the resulting uncertainties": this perspective is inherited from the companion paper. As will be indicated in my comments on that paper, it is not universally accepted and the problem is not so much with the concept per se as the fact that it is often
implemented by those without the skill set to do it properly. This needs to be reframed.

- Line 92: "can estimate" ⇒ "an estimate"?

- Line 93 "[classical flood frequency analysis] is not easily modified to allow for future change": I'm not sure that I agree with this, there are plenty of opportunities to incorporate nonstationarity into models of the underlying distributions — and, indeed, this is often done. I'm not sure that we should be promoting myths that originate with authors who are simply unaware of what is possible.

- Line 102 "Poisson distribution of occurrences": the GEV is a distribution of maxima, it has nothing to do with a Poisson distribution of occurrences. The Poissonness or otherwise of event occurrences therefore is irrelevant to whether or not the GEV is appropriate for modelling maxima (lines 107-108). The confusion here is with peaks-over-threshold approaches using the Generalised Pareto Distribution. This material would benefit from careful scrutiny by someone who is familiar with the relevant technical details of extreme value theory (there is at least one person on the author list who should have spotted this, which makes me wonder whether all of the authors have seen the manuscript?).

- Line 111: "less" ⇒ "fewer".

- Line 168 "epistemic uncertainty will remain a constraint on accuracy". This is true, but it would be helpful to articulate the implication which is that the best you can do is to ensure that your epistemic uncertainty is quantified and communicated clearly, so that users know at least the order of magnitude they can expect for discrepancies between what is predicted and what might occur.

- Line 203 "chance of flooding": does this mean "probability of flooding"? If so, change the legend: "uncertainty" is too imprecise a term. If it doesn't mean

"probability" (for example because it is simply a proportion of simulations) then this needs to be explained clearly, and there needs to be an acknowledgement that a proportion of simulations does not necessarily provide a decision-relevant probability that could be used in a formal risk assessment framework. This point has been emphasised repeatedly by more than one person, at every PURE consortium meeting where this kind of work has been presented.

- Line 282: "116 of which" $\Rightarrow$ "116 of whom".

- Line 500: delete "of" at end of line.

- Lines 527-530 "A particular feature of fitting such a stochastic model is that whether a model appears to give a good fit to the observed statistics might depend on the particular realisation ...": this seems to be the same point as lines 149-152, so there is an opportunity to lose a few lines. More generally, it is naive: the point is to assess whether the observed statistics (which are samples) could have been generated from the distributions implied by the simulations. There are people who have thought about this kind of thing rather carefully: their work should be represented.

- Line 557: delete "an" in "an upstream boundary conditions" (or make "conditions" singular)

- Line 846 "It has become established . . .": the jury is still out, isn't it? That is: if one considers aftershocks then recurrence is obviously non-Poisson, but the "quasi-cyclical" claim is not universally accepted I think. I note that the seismologist reviewer has serious concerns about the earthquake material in the paper and, in fact, recommends that the paper should be rejected. I don't particularly want to go that far, but the individual concerned knows what they are talking about and is not the kind of person to offer unfair criticism: the points made in his / her report require serious attention.

- Lines 1035-1036: the reported cost of the Eyja event seems very variable, depending on where you look! Wikipedia thinks that the cost to the *airline* industry was about 1.2 billion dollars in total, which seems inconsistent with the assertion that the total cost was a billion euros per day.

- Figure 3, caption "A negative BTD signal . . .": this is not clear to me, apart from anything else because there is no legend to tell us what the colour scale means.

- Line 1328: why *European* windstorms? Are issues of uncertainty different in Europe from elsewhere?!

- Line 1332: can you give a reference for the $200 billion figure? Is this in Europe or globally?

- Line 1368: why is "models" bold?

- Line 1393: this "wrap-up" section repeats material from the companion paper (e.g. in lines 1396-1400) so there is some scope for pruning back here to reduce the length.

- Line 1422: "exceed"⇒ "exceeded".

- Lines 1545-1546 "We hope that in making this comparison it will be researchers in different areas": something wrong here.

---

## Author Comment (AC1) · 31 Jul 2016

**Reviewer 1 comments:**

Overall, this section contains a useful description of the current uncertainty analysis being undertaken in the field of volcanic ash dispersion. However, the structure of the section is confusing in parts and several sources of epistemic uncertainty are missing from the description. Finally, explicit links to the uncertainty analysis types described in the paper's conclusions would help to link this section more closely to the overall aim of the paper. This is an ambitious and important paper that is worthy of publication.

Comments 1. Page 34, lines 6-11: The authors are rather dismissive about 'other types of observations'. Whilst there are obvious limitations associated with these low temporal or spatial resolution observations, they are often invaluable in measuring some of the physical properties of ash and ash cloud geometry that satellite retrievals currently need to make assumptions about in order to make concentration estimates. For example, ash layer depth, ash cloud height and particle size distribution. It might be better to acknowledge that combining these observations with the satellite observations can help to reduce the epistemic uncertainty?

We agree – this comes across as too brief and a line to explain the usefulness of other observations would be beneficial.

2. Page 33, line 18: I think 'in order to make estimates on other physical properties such as ash column loading ' should be ' in order to make estimates of other physical'

Corrected

3. Page 34, lines 3-5: The authors refer to 'simulations of satellite imagery using a radiative transfer model'. This is a bit ambiguous. I assume they are referring to volcanic ash satellite retrieval schemes, but another interpretation could refer to simulations of satellite imagery using VATD output, such as performed by Millington et al. (2015). This should be stated more clearly. More generally, 'the model' is also ambiguous as there are several models in this section. Millington, S. C., R. W. Saunders, P. N. Francis, and H. N. Webster (2012), Simulated volcanic ash imagery: A method to compare NAME ash concentration forecasts with SEVIRI imagery for the Eyjafjallajökull eruption in 2010, J. Geophys. Res., 117, D00U17, doi:10.1029/2011JD016770.

This is in reference to the volcanic ash retrieval schemes. Agreed that this should be made clearer by stating that, and elaborating on which "models" are being referred to.

4. Page 34, line 13: 'modelling of the hazard is, however, a problem of forecasting'. Since modelling in the previous 3 paragraphs was referring to volcanic ash satellite radiative transfer models, I was initially confused by this sentence. As the next few paragraphs go on to discuss VATD models, however, I assume modelling here refers to VATD models. It would be a good idea to clarify that from that point onwards the authors are discussing dispersion models.

Yes, this refers to VATD models. This has been clarified.

5. Page 34: When discussing parametric uncertainty, the authors focus almost exclusively on of the eruption source parameters (PSD, ash density, MER). I was surprised that uncertainty associated with the input meteorological fields was not discussed. Was this left out deliberately? If so, why?

Due to the length of the paper, we tried to keep the contribution short. However, brevity could be maintained by removing some discussion on the ESPs and adding more examples of where uncertainty originates.

6. Page 35, lines 5-18: The structural uncertainties associated with missing near source processes (bent plume, ash aggregation, effect of gravity currents) are dispersed through p34 and p35. Describing these missing processes in a separate paragraph might help with the clarity of the section.

Agreed, this section has been rearranged.

7. Page 35, line 20: The authors refer to the 'dispersion parameters' but there is no description of what these are, or any of the other parameter uncertainties associated with processes that are represented in VATD models, such as wet and dry deposition, sedimentation, turbulence etc. Why is there no discussion of parameter uncertainties?

Again, for the sake of brevity the deposition etc. parameters were not discussed, but is now included

8. Page 35: In the conclusions section the authors nicely split uncertainty analysis into 3 types. The parametric sensitivity studies described on page 35 all fall under the forward analysis approach. It might make it easier for the reader to link to the conclusions section if the authors could refer to the analysis type here.

Agreed – we have linked these back.

9. Page 36, lines 9-14. Similarly, how does the method of Denlinger et al. (2012) fit into the 3 types of uncertainty analysis framework? Is this a source inversion analysis?

This is an inversion type analysis, so could be moved next to the description of the Kristiansen et al. (2012) paper. But it is included because it uses an uncertainty analysis to estimate uncertainty in the forecast. We have revised this section.

10. Page 37, line 1-2. Given the operational time constraints, do the authors think that statistical emulators have a role to play in characterising volcanic ash forecasting uncertainties?

We know that this is being explored in the RACER research consortium, but we think this remains unclear.  As with any emulator, it will depend on how the predicted variables of far fewer runs of the full nonlinear model across the potential parameter space can be successfully interpolated, and perhaps conditioned on the available observations, to a posterior distribution.  In this case, reinitialisation for changing boundary conditions or meteorology would also require recalibration of the emulator. Dynamic emulation using systems theory model might also be considered but might also have limitations for these nonlinear problems where there is interaction between the

eruption and meteorology. These remain research questions of implementation, but are not necessarily relevant to the issues of epistemic uncertainty (except as so far as that might increase the dimensions to be considered).

**Comments from Reviewer 2**

We thank Reviewer 2 for the critical comments regarding the 'seismic hazards' section. In addressing the suggestions provided by Reviewer 2, we would like to clarify the main aim of the section. Later, we will explain how we revised the manuscript.

Firstly, in our manuscripts (both parts 1 and 2) we aimed at providing a general view on uncertainty assessment and modelling related to natural hazards. Our target audience should include both domain experts/scientists and practitioners/users. This is a challenging task given that, in effect, more in-depth coverage of individual topics is sought by some reviewers, where we are faced with a constraint on manuscript length (with some reviewers describing the manuscripts as too long, overall). In our revisions, we decided to take a more focused perspective (i.e. CREDIBLE view) on the uncertainty assessment and modelling related to natural hazards, rather than attempt an even more comprehensive review of this topic. In the revised manuscript, we state this purpose clearly in the Introduction.

As we aim at reaching a broader audience across various disciplines, we paid particular attention to providing a balanced view on the common themes. Because of this approach, discussions on the specific subjects are inevitably high level and incomplete. However, this allows a more commensurate perspective across several natural hazards.

We understand this reviewer's frustration from his/her specific comments which are mainly related to geological/tectonic aspects of seismic hazard. We agree that a thorough discussion on epistemic uncertainties related to geological/tectonic observations and parameters is very important. On the other hand, there are other important aspects in assessing seismic hazards and their uncertainties. We take a probabilistic seismic hazard analysis (PSHA) as a common framework (i.e. starting point) because it is the most widely implemented procedure (although there is on-going debate about the PSHA methodology in both scientific and applied communities). Thus, in the seismic hazard section, we have started with the general aspect of the treatment of epistemic uncertainties in current PSHA methodology (this is also useful for tsunami hazard, as probabilistic tsunami hazard analysis is gaining popularity), and then provided brief reviews on the specific model components and recent advances (e.g. in respect of earthquake occurrence modelling and ground motion modelling).

Below we try to address the reviewer's concerns and clarify some of the issues raised. In particular, to reduce length, a paragraph discussing the cascading earthquake hazard modelling is moved and integrated into Section 12.

**SPECIFIC COMMENTS**

Paragraph starting 815: This paragraph starts by introducing the problems of the long-term occurrence rates of extreme or "characteristic" earthquakes and then goes on to talk about maximum magnitudes. Further paragraph 846 talks about "non-Poisson and quasi-periodic" events at subduction zones. The language implicit asserts that large earthquake are at least semi-periodic since the language used talks about recurrence

rates and characteristic events. Both of these are debated issues which are, only potentially applicable to certain regions - only the latter paragraph makes this clear.

In the revised manuscript, we remove 'characteristic', and simply use 'large earthquakes'. We understand that the characteristic model is often adopted in certain special circumstances, e.g. where there is a major surface fault system where palaeoseismology can be attempted, and that the model doesn't have general applicability. As mentioned in the beginning of this reply, detailed critiquing of particular models is limited by the space available. Moreover, we did not intend to advocate one particular theory (i.e. quasi-periodic characteristic earthquake model). Rather, our intention was to note an extension of PSHA methodology with respect to the classical basis originated by Cornell (1968), which assumed a Poisson occurrence model and a basic seismic zone source configuration (compelled by computing limitations of the time).

Do the authors intend this discussion to be restricted to subduction zones or a wider coverage? For example, how do the Sichuan (Mw 7.9), New Madrid (8.1?), Gujarat (Mw 7.7) and Parkfield earthquakes fit into this framework?

Due to the limitation on manuscript length, we regret we are unable to extend the discussions to large intra-plate earthquakes, as suggested by the reviewer; while we agree strongly that characterising occurrence rates of these events is particularly challenging and involves considerable epistemic uncertainties, the issues cannot be treated sensibly and informatively in a few words.

Further, assuming a characteristic model is a potentially large source of epistemic uncertainty as there is not consensus about either the definition or validity of the characteristic model. Part of the challenge, is that records of seismic data are insufficiently short. An aleatoric reason for this is that the moments of the Gutenberg Richter relation are not finite. Measures of longer-term deformation are required to close this distribution. However, what will not resolve this issue (as implied by paragraph starting on line 815), is improving the completeness threshold or reliability of modern earthquake catalogues. Both of which are highly desirable, but will not contain the evidence to clinch long term recurrence rates of evidence for characteristic events, in my opinion. Hence, the second half of the paragraph does not address the 2 challenges identified in the first half.

We think that the reviewer over-interpreted the text in our manuscript; as mentioned above, we do not champion the characteristic model. On this issue, it seems to us that the reviewer and we share a similar view that inclusion of geological and geodetic information will help constrain seismic hazard estimates derived from PSHA. We have added a sentence at the end of this paragraph to make this clearer.

Paragraph starting 833: The inclusion of geodetic information will undoubtedly help refine estimates. Statistically, one of the problems with estimation of the maximum magnitude is that our evidence for it can only ever show us to have underestimated it. Since the geological process is one of the gradual destruction of evidence – we need to recognise large uncertainty in its evaluation. It would have been good to have shown how to consider these uncertainties within the geodetic case. Further, it seems unlikely we will be able to apply it globally and uniformly due to resource limitations. Since we generally go to look at regions we believe are scientifically interesting, we are biased at

looking at regions which have a signal. This bias means we can systematically miss swathes of hazardous regions.

We agree with the reviewer that including geodetic information in PSHA should be considered where it is possible, and that the geodetic information has considerable uncertainty. As part of the CREDIBLE consortium, this was explored in a PSHA study for Malawi (Hodge et al., 2015), using both instrumental catalogue data and geodetic information. We mention this work in text as a signpost so that interested readers can find further information. Related to this work, we had to face major challenges such as including a range of estimates of earthquake occurrence frequency along known faults and their rupture behaviour/pattern in terms of fault segmentation. Because these issues fall into characterising epistemic uncertainties in PSHA, in our revised manuscript we add two sentences (please see the revised manuscript – paragraph starting with 'Estimating frequency of occurrence …').

A good example of where we are developing new constraints is seafloor geodesy.

We agree that new insights from seafloor geodesy are very instructive. As part of the added text in the revision, we cite now a review paper by Bürgmann and Chadwell (2014).

INCOMPLETENESS OF DISCUSSION

There are also several examples of epistemic discussions missing from the paper. For example, how should one construct the background seismic hazard away from known active regions - Christchurch is a good example of this where the hazard was underestimated. Is there a prospect for PSHA to be able to inform those communities better in the future or was the earthquake and subsequent revision of building codes PSHA performing "well"? In a related matter, quantifying/mapping of what is measurable in principle also introduces systematic model uncertainties. Being sat on a modern sedimentary basin, the structure underlying the Christchurch region was masked. Similarly, a large proportion of earthquakes occur on unmapped structures. Geodetics can map slowly deforming structures – but not all unmapped structures are slowly deforming – they can remain locked until large earthquakes or similar change that situation. A further source of bias in the analysis of seismic hazard remains poor application of basic statistics. Many papers still apply least squares fitting to find the b-value – a process that was identified as being severely biased in the 1960s and yet is still widely applied. This application of using an inappropriate error structure and underestimating uncertainty at large magnitudes is certainly a classic example of applying the wrong model (i.e. epistemic).

As the reviewer points out, characterising the background seismicity in the context of PSHA is also important and challenging. Conventionally, this has been dealt with by developing a source zone model, often associated with annual occurrence rate and Gutenberg-Richter type magnitude distribution. Although it appears that epistemic uncertainties in relation to earthquake occurrence can be derived solely from data or catalogue information, our view is that this uncertainty quantification should be based on a wider analysis that also integrates expert judgment, to supplement the limitations on available data (which the reviewer notes). This applies equally to other factors and parameters in a PSHA, e.g. maximum magnitude.

In the revised manuscript, we have added one paragraph (starting with 'Characterising seismicity for the purposes of PSHA …'). In this paragraph, we mention the fundamental challenge of reliably quantifying epistemic uncertainties of background seismicity. This paragraph highlights the issues for characterising low-to-moderate diffused seismicity

for PSHA and, we hope, partly addresses some of the concerns mentioned by the reviewer, above.

**Editors Comments**

I was a little surprised to see that the Copernicus editorial system has automatically closed the discussion on this paper, because there are several other reviewers who haven't submitted their reports. I wanted a range of views from different hazard specialists, but only a couple of the areas are represented. Nonetheless, the comments received so far on this and on the companion paper are fairly consistent and in accordance with my own assessment, so it probably does no harm to return this one to the authors for a *very substantial* revision at this point. I'm afraid it needs a lot of work, if it is to fulfil its promise of a wide-ranging review in a special issue devoted to state-of-the-art treatment of uncertainty. Similar comments will come back on the companion paper in due course; I am still waiting for a final referee report on that one however, and I want to see that report before finalising my own comments.

The present paper sets out to review the ways in which generic issues of epistemic uncertainty, set out in the companion paper, have been addressed in different natural hazard areas. It inevitably inherits the perspective of the companion paper, so some of my major concerns about that paper carry through to this one as well. One of the most serious concerns is that both papers need to provide a much more balanced perspective: at present several techniques are criticised inappropriately, in a way that confuses the underlying concept with the way in which the techniques are often applied by non-specialists. I will say more about this in my report on the companion paper itself. However, for the present paper as well there are a few places where assertions need to be reframed to present a more balanced case.

An obvious general comment about the present paper is that it is very long. There is inevitably some overlap with the companion paper, but there are also a couple of places where essentially the same material is repeated and it would be good to identify opportunities to reduce the length.

A final general comment is that the treatment of the different hazard areas is rather uneven: for some areas the paper provides a reasonably comprehensive review of the landscape, but for others I have the sense that it does not so much provide a comprehensive review of the relevant issues as promote the relevant authors' own work. This is not really appropriate. The material on pyroclastic flows stands out particularly in this respect (although it is not the only offender): this reads less as a comprehensive review of uncertainty-relevant issues than as a list of gaps in our understanding of Vesuvius.

We fully understand the editor's perspective, but are sure he will appreciate the difficulty of getting a balance between length and being comprehensive across a wide range of natural hazard areas where the contributing experts have not all needed to think in terms of epistemic uncertainty issues before.  That is where the approach is different from the Rougier et al. book (which was also of course much much longer).  So as in Paper 1 the intention was to be wide-ranging but not necessarily comprehensive.  This is also one reason why, in working from personal experience of dealing with such

problems, the co-authors have tended (understandably) to cite their own work. This has now been made much clearer in the introduction.

In revising the paper we have tried to be much more explicit about referring to the controversies and open questions discussed in Paper 1 while trying to avoid the repetition noted in this version. Following the complete revision of Paper 1, we have also tried to distinguish between where appropriate statistical concepts might apply, where they might be poorly applied by non-specialists, and where they might still have important limitations in dealing with epistemic uncertainties (such as those gaps in our understanding of Vesuvius and many other examples). All the co-authors have been asked to consider both length and their citations and particularly self-citations.

All the sections of the paper have been revised accordingly.

We have also addressed the specific comments that follow in the revised manuscript (where they are still relevant) and thank the editor (again) for taking the time and care to comment in detail.

SPECIFIC COMMENTS

Lines 25–28 "In each case it is common practice . . . underestimation of the resulting uncertainties": this perspective is inherited from the companion paper. As will be indicated in my comments on that paper, it is not universally accepted and the problem is not so much with the concept per se as the fact that it is often implemented by those without the skill set to do it properly. This needs to be reframed.

See the response to the general comments.

Line 92: "can estimate" ⇒ "an estimate"?

Reworded

Line 93 "[classical flood frequency analysis] is not easily modified to allow for future change": I'm not sure that I agree with this, there are plenty of opportunities to incorporate nonstationarity into models of the underlying distributions — and, indeed, this is often done. I'm not sure that we should be promoting myths that originate with authors who are simply unaware of what is possible.

We are only too aware of what is possible and what is being done (generally badly). Models of future change generally have poor hydrology and do not produce flood estimates directly. But the results are not generally presented as conditional on the assumptions made. And there are limitations due to epistemic uncertainties even for what might be considered as (conditional) good practice.

Line 102 "Poisson distribution of occurrences": the GEV is a distribution of maxima, it has nothing to do with a Poisson distribution of occurrences. The Poissonness or otherwise of event occurrences therefore is irrelevant to whether or not the GEV is appropriate for modelling maxima (lines 107-108). The confusion here is with peaks-over-threshold approaches using the Generalised Pareto Distribution. This material would benefit from careful scrutiny by someone who is familiar with the relevant technical details of extreme value theory (there is at least one person on the author list who should have spotted this, which makes me wonder whether all of the authors have seen the manuscript?).

This has now been rewritten

Line 111: "less" ⇒ "fewer".

Corrected

Line 168 "epistemic uncertainty will remain a constraint on accuracy". This is true, but it would be helpful to articulate the implication which is that the best you can do is to ensure that your epistemic uncertainty is quantified and communicated clearly, so that users know at least the order of magnitude they can expect for discrepancies between what is predicted and what might occur.

But that is exactly the point we are trying to make -

Line 203 "chance of flooding": does this mean "probability of flooding"? If so, change the legend: "uncertainty" is too imprecise a term. If it doesn't mean  "probability" (for example because it is simply a proportion of simulations) then this needs to be explained clearly, and there needs to be an acknowledgement that a proportion of simulations does not necessarily provide a decision-relevant probability that could be used in a formal risk assessment framework. This point has been emphasised repeatedly by more than one person, at every PURE con- sortium meeting where this kind of work has been presented.

Line 282: "116 of which" ⇒ "116 of whom".

Corrected

Line 500: delete "of" at end of line.

Corrected

Lines 527-530 "A particular feature of fitting such a stochastic model is that whether a model appears to give a good fit to the observed statistics might depend on the particular realisation ...": this seems to be the same point as lines 149-152, so there is an opportunity to lose a few lines. More generally, it is naive: the point is to assess whether the observed statistics (which are samples) could have been generated from the distributions implied by the simulations. There are people who have thought about this kind of thing rather carefully: their work should be represented.

Now rewritten

Line 557: delete "an" in "an upstream boundary conditions" (or make "conditions" singular)

Corrected

Line 846 "It has become established . . .": the jury is still out, isn't it? That is: if one considers aftershocks then recurrence is obviously non-Poisson, but the "quasi- cyclical" claim is not universally accepted I think. I note that the seismologist reviewer has serious concerns about the earthquake material in the paper and, in fact, recommends that the paper should be rejected. I don't particularly want to go that far, but the

individual concerned knows what they are talking about and is not the kind of person to offer unfair criticism: the points made in his / her report require serious attention.

See response to reviewer 2

Lines 1035-1036: the reported cost of the Eyja event seems very variable, depending on where you look! Wikipedia thinks that the cost to the airline industry was about 1.2 billion dollars in total, which seems inconsistent with the assertion that the total cost was a billion euros per day.

This figure refers to the total economic cost of each day air space was closed, not just the cost to airlines. It is changed to make it less ambiguous, e.g.: "The total global cost in GDP over the entire eruption was $5 billion (Oxford economics, 2010)."

Figure 3, caption "A negative BTD signal" this is not clear to me, apart from anything else because there is no legend to tell us what the colour scale means

The legend on the figure gives the negative brightness temperature difference in Kelvin. The greyscale is the positive BTD. The figure caption has been updated.

"Figure 3 Meteosat Second Generation Spinning Enhanced Visible and InfraRed Imager (SEVIRI) brightness temperature difference image (brightness temperature at the 10.8 μm channel minus the brightness temperature at the 12 μm channel) indicating the extent of the Eyjafjallajökull ash cloud at 0300 UTC 8th May 2010. The negative values of BTD (indicating ash) are shown in blue and the scale in Kelvin is given on the legend. The positive BTD is plotted in grey. A likely false negative ash signal can be seen south of Iceland where the ash plume appears to be obscured, possibly by meteorological cloud, due to the high ash concentration causing opaqueness, a large fraction of large particles or the presence of water in the plume. A negative BTD signal can be seen over North Africa / south eastern Spain, possibly due to a night-time clear arid land surface. Raw data supplied by EUMETSAT."

Line 1328: why *European* windstorms? Are issues of uncertainty different in Europe from elsewhere?!

Rewritten

Line 1332: can you give a reference for the $200 billion figure? Is this in Europe or globally?

Corrected

Line 1368: why is "models" bold?

Corrected

Line 1393: this "wrap-up" section repeats material from the companion paper (e.g. in lines 1396-1400) so there is some scope for pruning back here to reduce the length.

Now rewritten

Line 1422: "exceed"⇒ "exceeded".

Corrected

Lines 1545-1546 "We hope that in making this comparison it will be researchers in different areas": something wrong here.

Corrected

**Additional comment**

The comment by Stefano Luigi Gariano is very positive, and suggests for us to include a paper for which he is the main author. This is relevant and has now been included.

Reference:
Gariano S.L., Brunetti M.T., Iovine G., Melillo M., Peruccacci S., Terranova O.G., Vennari C., Guzzetti F. (2015), Calibration and validation of rainfall thresholds for shallow landslide forecating in Sicily, Southern Italy, Geomorphology, 228, 653-665, doi:10.1007/s11069-014-1129-0.